# TIME-DEPENDENT REPRESENTATION FOR NEURAL EVENT SEQUENCE PREDICTION

## ABSTRACT

Existing sequence prediction methods are mostly concerned with time-independent sequences, in which the actual time span between events is irrelevant and the distance between events is simply the difference between their order positions in the sequence. While this time-independent view of sequences is applicable for data such as natural languages, e.g., dealing with words in a sentence, it is inappropriate and inefficient for many real world events that are observed and collected at unequally spaced points of time as they naturally arise, e.g., when a person goes to a grocery store or makes a phone call. The time span between events can carry important information about the sequence dependence of human behaviors. In this work, we propose a set of methods for using time in sequence prediction. Because neural sequence models such as RNN are more amenable for handling token-like input, we propose two methods for time-dependent event representation, based on the intuition on how time is tokenized in everyday life and previous work on embedding contextualization. We also introduce two methods for using next event duration as regularization for training a sequence prediction model. We discuss these methods based on recurrent neural nets. We evaluate these methods as well as baseline models on five datasets that resemble a variety of sequence prediction tasks. The experiments revealed that the proposed methods offer accuracy gain over baseline models in a range of settings.

## 1 INTRODUCTION

Event sequence prediction is a task to predict the next event[1] based on a sequence of previously occurred events. Event sequence prediction has a broad range of applications, e.g., next word prediction in language modeling (Józefowicz et al., 2016), next place prediction based on the previously visited places, or next app to launch given the usage history. Depending on how the temporal information is modeled, event sequence prediction often decomposes into the following two categories: discrete-time event sequence prediction and continuous-time event sequence prediction.

Discrete-time event sequence prediction primarily deals with sequences that consist of a series of tokens (events) where each token can be indexed by its order position in the sequence. Thus such a sequence evolves synchronously in natural unit-time steps. These sequences are either inherently time-independent, e.g, each word in a sentence, or resulted from sampling a sequential behavior at an equally-spaced point in time, e.g., busy or not busy for an hourly traffic update. In a discrete-time event sequence, the distance between events is measured as the difference of their order positions. As a consequence, for discrete-time event sequence modeling, the primary goal is to predict what event will happen next.

Continuous-time event sequence prediction mainly attends to the sequences where the events occur asynchronously. For example, the time interval between consecutive clinical visits of a patient may potentially vary largely. The duration between consecutive log-in events into an online service can change from time to time. Therefore, one primary goal of continuous-time event sequence prediction is to predict when the next event will happen in the near future.

---

[1] We use "event" in this paper for real world observations instead of "token" that is often used in sequence problems, e.g., words in a sentence. But they are equivalent to a sequence model.

Although these two tasks focus on different aspects of a future event, how to learn a proper representation for the temporal information in the past is crucial to both of them. More specifically, even though for a few discrete-time event sequence prediction tasks (e.g., neural machine translation), they do not involve an explicit temporal information for each event (token), a proper representation of the position in the sequence is still of great importance, not to mention the more general cases where each event is particularly associated with a timestamp. For example, the next destination people want to go to often depends on what other places they have gone to and how long they have stayed in each place in the past. When the next clinical visit (Choi et al., 2016a) will occur for a patient depends on the time of the most recent visits and the respective duration between them. Therefore, the temporal information of events and the interval between them are crucial to the event sequence prediction in general. However, how to effectively use and represent time in sequence prediction still largely remains under explored.

A natural and straightforward solution is to bring time as an additional input into an existing sequence model (e.g., recurrent neural networks). However, it is notoriously challenging for recurrent neural networks to directly handle continuous input that has a wide value range, as what is shown in our experiments. Alternatively, we are inspired by the fact that humans are very good at characterizing time span as high-level concepts. For example, we would say "watching TV for a little while" instead of using the exact minutes and seconds to describe the duration. We also notice that these high-level descriptions about time are event dependent. For example, watching movies for 30 minutes might feel much shorter than waiting in the line for the same amount of time. Thus, it is desirable to learn and incorporate these time-dependent event representations in general. Our paper offers the following contributions:

- We propose two methods for *time-dependent event representation* in a neural sequence prediction model: time masking of event embedding and event-time joint embedding. We use the time span associated with an event to better characterize the event by manipulating its embedding to give a recurrent model additional resolving power for sequence prediction.

- We propose to use *next event duration as a regularizer* for training a recurrent sequence prediction model. Specifically, we define two flavors of duration-based regularization: one is based on the negative log likelihood of duration prediction error and the other measures the cross entropy loss of duration prediction in a projected categorical space.

- We evaluated these proposed methods as well as several baseline methods on five datasets (four are public). These datasets span a diverse range of sequence behaviors, including mobile app usage, song listening pattern, and medical history. The baseline methods include vanilla RNN models and those found in the recent literature. These experiments offer valuable findings about how these methods improve prediction accuracy in a variety of settings.

## 2 BACKGROUND

In recent years, recurrent neural networks (RNN) especially with Long-Short Term Memory (LSTM) (Hochreiter & Schmidhuber, 1997) have become popular in solving a variety of discrete-time event sequence prediction problems, including neural machine translation (Bahdanau et al., 2014), image captioning (Xu et al., 2015) and speech recognition (Soltau et al., 2016). In a nutshell, given the sequence of previously occurred events, $\{e_1, e_2, ..., e_t\}$, the conditional probability $P(e_{t+1}|\{e_1, e_2, ..., e_t\}) = P(e_{t+1}|h_t, \theta)$ of the next event $e_{t+1}$ is estimated by using a recurrent neural network with parameters $\theta$ and the hidden state vector $h_t = f(h_{t-1}, e_t, \theta)$ which is assumed to encode the information of the past events.

To feed an event into a recurrent neural network, the event, often described as a categorical variable, needs to be represented in a continuous vector space. A common way to achieve this is to use embedding (Bengio et al., 2003) $x_t = 1(e_t)E^x$ where $1(e_t)$ is a one-hot vector. For the $j$th event in the vocabulary $V$, $e^j$, its one-hot vector has 0s for all the entries except the $j$th entry being 1. $E^x \in R^{|V| \times E}$ is the embedding matrix, where $|V|$ is the number of unique events (the vocabulary size) and $E$ is the embedding dimension. The use of embedding provides a dense representation for an event that improves learning (Turian et al., 2010). Through training, the embedding vector of an event encodes its meaning relative to other events. Events that are similar tend to have embedding vectors closer to each other in the embedding space than those that are not.

On the other hand, temporal point processes are mathematical abstractions for the continuous-time event sequence prediction task by explicitly modeling the inter-event interval as a continuous random variable. Since the occurrence of an event may be triggered by what happened in the past, we can essentially specify different models for the timing of the next event given what we have already known so far. Very recently, (Du et al., 2016; Mei & Eisner, 2017; Xiao et al., 2017a;b) focus on expanding the flexibility of temporal point processes using recurrent neural networks where the prediction of the next event time is based on the current hidden state $h_t$ of RNN. However, all of these work use the direct concatenation between the inter-event interval and the respective event embedding as the input to the recurrent layer where the representation of the temporal information is limited.

Because it is not clear how to properly represent time as input, in this work, we intend to let the model learn a proper representation for encoding temporal information in a sequence, similar to learning embeddings for words. Rather than proposing a new model, our approach should be considered an "embedding" approach for time that can be used by general event sequence prediction models, including models proposed previously (Du et al., 2016; Mei & Eisner, 2017).

## 3 TIME-DEPENDENT EVENT REPRESENTATION

There are two notions about time spans in a sequential behavior: duration and intervals. Duration is how long an event lasts, e.g., listening to music for an half hour, and an interval is the time span between two adjacent events. To unify both types of time spans, we treat the idle period when no event is occurring (e.g., the person is not using any app for an app usage sequence) as a special event. Thus, duration becomes an inherent property of an event–the interval between two events is the duration of an idle event (see Figure 1).

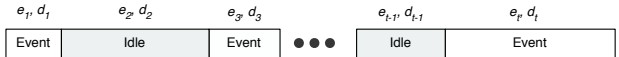

Figure 1: An interval is treated as the "duration" of an idle event.

With this, $h_t = f(h_{t-1}, e_t, d_t; \theta)$ where $d_t$ is the duration of event $e_t$. We here propose two methods to bring continuous time, $d_t$, into a neural sequence prediction model. Both achieve time-dependent event representation by manipulating event embedding vectors using time. Our methods are schematically illustrated in Figure 2.

### 3.1 CONTEXTUALIZING EVENT EMBEDDING WITH TIME MASK

Recent work by (Choi et al., 2016b) revealed that in neural machine translation the embedding vector of a word encodes multiple meanings of the word. As a result, it requires a recurrent layer to sacrifice its capacity to disambiguate a word based on its context, instead of focusing on its main task for learning the higher-level compositional structure of a sentence. To address this problem, they used a mask computed based on all the words in a sentence to contextualize the embedding of a target word.

Based on this recent work, we propose a method to learn a time mask to "contextualize" event embedding, by which we hope a time-dependent embedding would give the recurrent layer additional resolving power. Similar to the word mask proposed by Choi et al. (Choi et al., 2016b), we first compute a time context vector for duration, $c^d$.

$$c^d = \phi(\log(d_t); \theta) \tag{1}$$

$\phi$ is a nonlinear transformation of $d_t$ and is implemented as a feedforward neural network parameterized by $\theta$. $d_t$ is log transformed before it is fed to $\phi$ to effectively cover the wide numerical range of duration values, e.g., it can range from seconds to hours for app usage events.

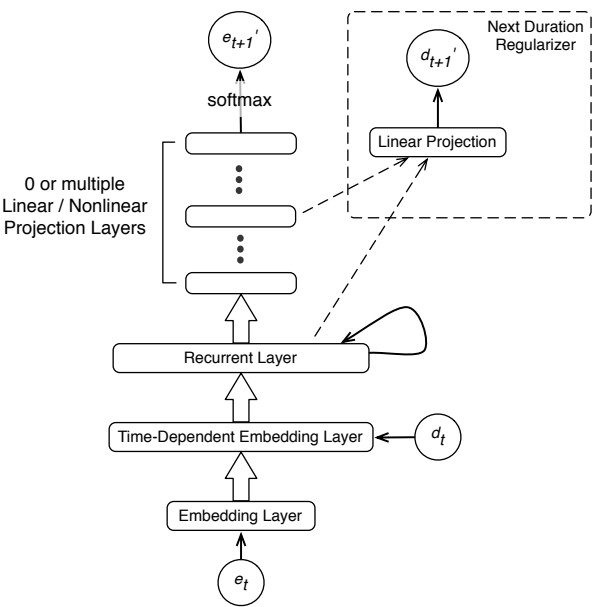

Figure 2: A time-dependent RNN for event sequence prediction. $d_t$ is used to generate time-dependent event embedding. Next event duration can be used as a regularizer, which can be applied to the recurrent layer and/or any post recurrent layer.

We compute a time mask by linearly transforming $c^d$ with weights $W_d \in \mathbb{R}^{C \times E}$ and bias $b_d \in \mathbb{R}^E$, which is followed by a sigmoid nonlinear activation, $\sigma$, to generate a mask $m_d \in \mathbb{R}^E$ and $\mathbb{R}^E \to [0, 1]$. $C$ is the size of the time context vector, and $E$ is the event embedding dimension.

$$m_d = \sigma(c^d W_d + b_d) \tag{2}$$

We then apply the mask to an event embedding by performing an element-wise multiplication, $\odot$, between the embedding vector and the mask. Finally, the product is fed to the recurrent layer.

$$x_t \leftarrow x_t \odot m_d \tag{3}$$

## 3.2 EVENT-TIME JOINT EMBEDDING

Humans developed many ways to tokenize continuous time in everyday life. For example, we would say "talk to someone briefly" instead of using exact minutes and seconds to characterize the length of the conversation. Such a kind of tokenization is extensively used in natural languages. In addition, our perception about the duration also depends on the specific event that we are experiencing. Based on these intuitions, we propose a method to first encode the duration of an event using *soft* one-hot encoding and then use the encoding to form the joint embedding with the event.

To do so, we first project the scalar duration value onto a vector space, where $W_d \in \mathbb{R}^{1 \times P}$ is the weight matrix, $b_d \in \mathbb{R}^P$ is the bias vector, and $P$ is the projection size.

$$p^d = d_t W_d + b_d \tag{4}$$

We then compute the soft one-hot encoding, $s^d$, of a duration value by applying a softmax function to the projection vector, $p^d$. Softmax has been typically used in the output layer (Graves, 2012) and in

the attention mechanisms (Bahdanau et al., 2014; Xu et al., 2015) for selecting one out of many. The $i$th entry of the encoding vector is calculated as the following and $p_i^d$ is the $i$th entry in $p^d$.

$$s_i^d = \frac{\exp(p_i^d)}{\sum_{k=1}^{P} \exp(p_k^d)} \tag{5}$$

All the entries in the soft one-hot encoding are positive. Similar to a regular one-hot encoding, $\sum_{i=1}^{P} s_i^d = 1$. We then project the soft one-hot encoding onto a time embedding space, $g_d$. It has the same dimension as the event embedding. $E^s \in R^{P \times E}$ is the embedding matrix.

$$g_d = s^d E^s \tag{6}$$

Embedding for a regular one-hot encoding essentially takes a single row of the embedding matrix that is corresponding to the non-zero entry as the embedding vector. In contrast, embedding for a soft one-hot encoding computes a weighted sum over all the rows in the embedding matrix. Finally, we form the joint embedding of an event and its duration by taking the mean of their embedding vectors, which is then fed to the recurrent layer.

$$x_t \leftarrow \frac{x_t + g_d}{2} \tag{7}$$

## 4 NEXT EVENT DURATION AS A REGULARIZER

While our goal here is to predict next event, it can help learning by introducing an additional loss component based on the prediction of the next event duration (see Figure 2). The duration prediction of the next event at step $t$, $d'_{t+1}$, is computed from a linear transformation of the recurrent layer. A loss defined on the prediction error of $d'_{t+1}$ provides additional information during back propagation, acting like a regularizer. Optionally, one can use the concatenation of the recurrent layer output and a hidden layer on the path for event prediction to regularize more layers. We discuss two alternatives for the loss function over $d'_{t+1}$.

### 4.1 NEGATIVE LOG LIKELIHOOD OF TIME PREDICTION ERROR

A common way for the loss over a continuous value is to use the squared error. Here, it is $(d'_{t+1} - d_{t+1})^2$ where $d_{t+1}$ is the observed duration of the next event. However, such a loss needs to be at the same scale as that of of event prediction, which is typically a log likelihood of some form. Hinton and Van Camp (Hinton & van Camp, 1993) have shown that minimizing the squared error can be in fact formulated as maximizing the probability density of a zero-mean Gaussian distribution. Note that this does not require duration to obey a Gaussian distribution but rather the prediction error. We define our regularizer, $R_t^N$, as the negative log likelihood of duration prediction error at step $t$.

$$R_t^N = \frac{(d'_{t+1} - d_{t+1})^2}{2\sigma_i^2} \tag{8}$$

The variance, $\sigma_i$, is seeded with an initial value (e.g., the variance of duration values in the training data) and updated iteratively during training based on the duration prediction error distribution of the learned model at each update $i$.

### 4.2 CROSS ENTROPY LOSS ON TIME PROJECTION

In Section 3.2, we proposed to use softmax to project a continuous duration value onto a categorical space. Using the same technique, by projecting both $d'_{t+1}$ and $d_{t+1}$ onto a categorical space, we can then compute a cross entropy loss based on the two projections as another regularizer $R_t^X$.

$$R_t^X = -\sum_{k=1}^{P} Proj_k(d_{t+1}) \log Proj_k(d'_{t+1}) \tag{9}$$

$Proj$ is the softmax projection process we defined in Equation 4 and 5, $Proj_k$ is the $k$th entry in the projection vector. When event-time joint embedding and $R_t^X$ are both used, the embedding and the regularizer can use the same projection function, i.e., sharing the same projection weights (Equation 4).

## 5 EXPERIMENTS

In this section, we evaluate the effectiveness of our proposed approaches on the following five real-world datasets across a diverse range of domains.

- Electrical Medical Records. MIMIC II medical dataset is a collection of de-identified clinical visit records of Intensive Care Unit patients for seven years. The filtered dataset released by (Du et al., 2016) include 650 patients and 204 diseases. The goal is to predict which major disease will happen to a given patient.

- Stack Overflow Dataset. The Stack Overflow dataset includes two years of user awards on a question-answering website. The awarded badges are treated as the events. (Du et al., 2016) collected 6,000 users with a total of 480,000 events. The goal is to predict the next badge a user will receive.

- Financial Transaction Dataset. (Du et al., 2016) collected a long stream of high frequency transactions for a single stock from NYSE where the events correspond to the "buy" and "sell" actions. The task is to predict the next action a user might take.

- App Usage Dataset. Mobile users often use a large number of apps, ranging from tens to hundreds. It is time consuming to find a target app on mobile devices. One promising way to address this problem is to predict the next app a user will use based on their app usage history. Being able to predict next apps also allows the mobile platform to preload an app in memory to speed up its startup. We have collected 5,891 app usage sequences comprising of 2.8 million app usage events. The task is to predict the next app that will be used for a given user.

- Music Recommendation. The music dataset represents the longitudinal listening habits of 992 users (Last.FM, 2009; Celma, 2010) involving millions of listening events. The goal is to predict the next five unique songs that the user has not listened given the user's listen history.

### 5.1 DATA PREPARATION

For the MIMIC II, Stack Overflow, and Financial data, we follow (Du et al., 2016) to pre-process the data and seek to predict every single held-out event from the history. We evaluate the prediction accuracy with the binary 0-1 loss.

For the app usage data, to avoid users who participated in the data collection only briefly, we exclude sequences that have fewer than 50 app launches or if the time span of the sequence is shorter than a week. This resulted in 5,891 app usage sequences, one from each unique user. These sequences include 2,863,095 app usage events and the longest sequence spanned 551 days. We split the dataset on users into the training (80%), validation (10%) and test (10%) such that each user is only in one of these partitions. Hence there is no intersection of users between training, validation and test sets. For an event that has fewer than 5 occurrences in the training dataset, we assign it the OOV id for out of vocabulary. In total, there are 7,327 events in the vocabulary, including 7,325 unique apps, the idle event and the OOV (out of vocabulary). In practice, predicting the next 5 apps is often desired so we use Precision@K to evaluate the performance.

For the music recommendation, each listen event has a timestamp. We removed sequences that are shorter than 50 and songs that have fewer than 50 listens. We thus generate a collection of examples where each example consists of a listen history and a set of 5 unique songs to recommend. To do so, we split each original listen sequence into segments. We first take the 40 events out in order from the beginning of the sequence as the listen history, and then take more events out from the beginning of the sequence until we find 5 unique songs that have not occurred in the listen history. We do so repeatedly to extract each example until we exhaust all the original sequences. This data processing resulted in 221,920 sequence examples with 71,619 unique songs (the vocabulary size). We then

allocate these sequence examples for the training (80%), validation (10%) and test (10%). Because the original dataset does not have the duration information for each listen event, we did not inject the additional idle event in the sequence to differentiate duration versus intervals. Because in practice, the ranking order of the recommended music often matters, we further use MAP@K and Prevision@K to evaluate the performance.

## 5.2 Model Configurations

We compare with the following five models: *NoTime* in which a simple LSTM sequence model is used; *TimeConcat* in which we feed time (log transformed) directly into the recurrent layer along the event embedding; *TimeMask* (Section 3.1) and *TimeJoint* (Section 3.2) for generating time-dependent event embedding as input to the recurrent layer; and *RMTPP* for the model introduced previously by (Du et al., 2016). Moreover, we also include four regularized models based on $R_t^X$ and $R_t^N$ defined earlier. For *TimeMask*, the size of the time context vector is $C = 32$, and we use ReLu for the activation function in $\phi$ in Equation 2. For *TimeJoint*, we chose the projection size, $P = 30$ (Equation 4). For the App Usage and Music Recommendation experiments, we use a two-layer hierarchical softmax (Morin & Bengio, 2005) for the output layer due to the large vocabulary size, while we use a full sofmax for the rest experiments.

For the MIMIC II, Stack Overflow, and Financial data, we follow (Du et al., 2016) for *RMTPP*'s model parameters. For the app usage data, we determined the parameters of each model based on the training and the validation datasets on a distributed parallel tuning infrastructure. We used LSTM units (Hochreiter & Schmidhuber, 1997) for the recurrent layer, and Rectified Linear Units (ReLu) (Nair & Hinton, 2010) for the activation function in the nonlinear projection layer. The event embedding dimension, the number of LSTM units, and the nonlinear projection layer size are all set to 128. For the music recommendation data, we use a setting similar to the app prediction experiment where we chose the embedding size as 128 and LSTM size as 256. We did not use the nonlinear projection layer after the LSTM layer for this task because it does not seem to help. We implemented all the models in TensorFlow (TensorFlow, 2017).

## 5.3 Training and Testing

For the experiments based on MIMIC II, Stack Overflow and Financial Transaction datasets, we use the same training and testing strategy of (Du et al., 2016). For App Usage and Music Recommendation tasks, we selected the model architecture and hyper parameters with early stopping based on the validation dataset of each task, and report the performance of each model based on the test dataset.

For the App Usage experiment, we used truncated back-propagation through time with the number of unroll to be 30. We used an adaptive gradient descent optimizer (Zeiler, 2012), using a learning rate of 0.024 with a threshold for gradient clipping of 1.0, and a batch size of 32. We decided not to use dropout as it did not seem to improve accuracy on this task.

For the Music Recommendation experiment, we used the full sequence back-propagation through time with 2% dropout ratio on the recurrent layer for better generalization. We used the Adam optimizer by (Kingma & Ba, 2014) for adaptive learning with a learning rate of 0.00005 and a gradient clipping threshold at 1.0. The mini-batch size is 256.

We trained the models by minimizing the cross-entropy loss, plus the regularization loss if the duration regularizer is used, over all the sequences in the training dataset. The training for App Usage and Music Recommendation was conducted on a distributed learning infrastructure (Dean et al., 2012) with 50 GPU cores where updates are applied asynchronously across multiple replicas.

## 5.4 Experimental Results

*Effectiveness of Temporal Representation*. Figure 3 presents the comparisons between all the models on three released public datasets. We can observe a consistent performance gain with using the proposed methods for time-dependent event embedding compared to the *NoTime* baseline and the simple *TimeConcat* approach. *TimeJoint* significantly outperformed all other methods on both the Stack Overflow and the Financial dataset, with p<0.05 using Paired T-test. But none of the methods for using time is able to improve accuracy on the MIMIC II dataset. This indicates that using time

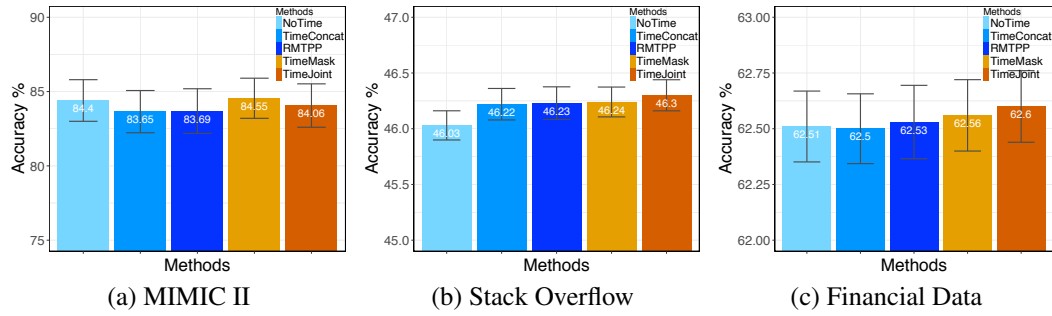

Figure 3: Prediction accuracy on (a) MIMIC II, (b) Stack Overflow, and (c) Financial Data.

Table 1: Prediction accuracy on test dataset for next app prediction in percentages.

| Model | Precision@1 | Precision@5 |
|---|---|---|
| NoTime | 30.29 | 13.07 |
| TimeConcat | 31.03 | 12.98 |
| RMTPP | 31.31 | 12.9 |
| TimeMask | 31.29 | 13.13 |
| TimeMask + $R_t^X$ | 31.3 | **13.15** |
| TimeMask + $R_t^N$ | 31.41 | 13.1 |
| TimeJoint | 31.3 | 13.07 |
| TimeJoint + $R_t^X$ | **31.53** | 13.09 |
| TimeJoint + $R_t^N$ | 31.45 | 13.13 |

Table 2: Prediction accuracy on test dataset for music recommendation. The numbers are percentages.

| Model | MAP5 | MAP10 | MAP20 | Precision@5 | Precision@10 | Precision@20 |
|---|---|---|---|---|---|---|
| NoTime | 11.59 | 13.18 | 13.83 | 13.82 | 8.75 | 5.25 |
| TimeConcat | 11.41 | 12.85 | 13.51 | 13.53 | 8.63 | 5.19 |
| RMTPP | 11.51 | 12.93 | 13.59 | 13.62 | 8.66 | 5.19 |
| TimeMask | 11.74 | 13.18 | 13.83 | 13.79 | 8.81 | 5.28 |
| TimeMask + $R_t^X$ | 11.71 | 13.17 | 13.82 | 13.81 | 8.76 | 5.25 |
| TimeMask + $R_t^N$ | 11.69 | 13.16 | 13.8 | 13.81 | 8.76 | 5.30 |
| TimeJoint | 11.82 | 13.37 | 14.06 | 13.95 | 8.97 | 5.40 |
| TimeJoint + $R_t^X$ | **12.02** | **13.51** | **14.2** | **14.12** | **9.01** | **5.43** |
| TimeJoint + $R_t^N$ | 11.9 | 13.41 | 14.11 | 14.05 | 8.98 | 5.40 |

might not always help. However, when it does, our methods such as *TimeJoint* enable more efficient representation of time than simply using the scalar value of time in RNN models.

Our methods also outperformed *RMTPP* for event prediction. The performance gain of our models are more pronounced on the App Usage and Music Recommendation datasets as shown in Table 1 and 2. *TimeJoint* seems to outperform the rest on most measures and *TimeMask* also performs well compared to other previous methods. We also notice that using time directly without representing them appropriately in RNN, i.e., *TimeConcat*, can sometime hurt the performance.

*Effectiveness of Event Duration Regularization*. We demonstrate the performance boosting gained from our proposed temporal regularization in Table 1 and 2, respectively. We can observe that our proposed regularizers can bring additional performance gain on many cases. In particular, the cross-entropy regularizer, $R_t^X$, is able to give consistent performance gain with the temporal embedding approaches.

*Learned Time Representation*. Our motivation in this work is to let the model learn a proper representation of time from data. We here briefly discuss what the TimeJoint approach learns about

how to project a scalar value of time into a soft one-hot encoding 4. It seems that for small time periods, e.g., shorter than 20 seconds for the Next App prediction task, more dimensions are needed to express the differences of continuous time values. As the time period grows, we need less dimensions for representing time, e.g., two of the curves have converged to the same small values.

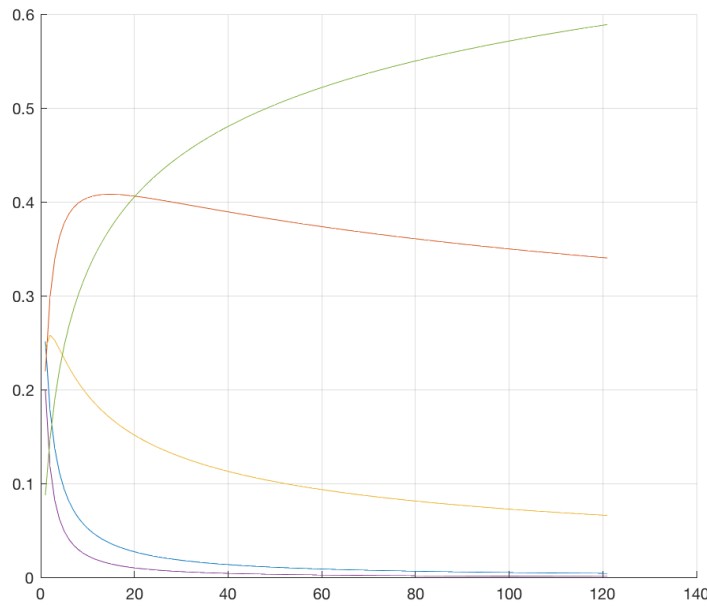

Figure 4: The projection of time learned by TimeJoint with $P = 5$. The X axis is in seconds and the Y axis is the projection of a time in each dimension defined in Equation 5.

## 6 CONCLUSIONS

We proposed a set of methods for leveraging the temporal information for event sequence prediction. Based on our intuition about how humans tokenize time spans as well as previous work on contextual representation of words, we proposed two methods for time-dependent event representation. They transform a regular event embedding with learned time masking and form time-event joint embedding based on learned soft one-hot encoding. We also introduced two methods for using next duration as a way of regularization for training a sequence prediction model. Experiments on a diverse range of real data demonstrate consistent performance gain by blending time into the event representation before it is fed to a recurrent neural network.

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
