# OpenReview forum: "Time-Dependent Representation for Neural Event Sequence Prediction"
_ICLR.cc/2018/Conference — Invite to Workshop Track_

### Official Review · AnonReviewer1 · 2017-11-12
**An interesting attempt for time-event information fusion, but can be much improved.**

**Rating:** 4
**Confidence:** 4

**Review:**

Quality above threshold.
Clarity above threshold.
Originality slightly below threshold.
Significance slightly below threshold.

Pros:
This paper proposed a RNN for event sequence prediction. It provides two constructed choices for combining time(duration) information to event. Experiments on various datasets were conducted and most details are provided.

Cons (concerns):

1. Event sequence prediction is a hard problem as there’s no clear way to fuse the features about event and the information about the time. It is a nice attempt that in this work, duration is used for event representation. However, the choices are not “principled” as claimed in the paper. E.g., the duration is simply a scaler, but "time mask" approache converts that to a multi-dimensional vector while there’s not much information to regularize it.

2. Event-time joint embedding sounds sensible as it essentially remaps from the original value to some segments. E.g., 10 minutes and 11 minutes might have same effect on next event in one dataset while 3 days and a week might have similar effect on next event prediction. But the way how the experiments are designed and analyzed do not provide such insights.

3. The experimental results are not persuasive as no other baselines besides RNN-based methods are provided. Parametric and nonparametric methods both exist for this event prediction problem in previous work. In the results provided, no significant difference between the listed model choices is found, partly because only using event type and duration is not enough. Other info such as time of day, day of week matters a lot.

---

> ### Author Response · Authors · 2018-01-05
> **Responses to Reviewer1's comments**
>
> Regarding the baseline methods, previous work (e.g., Du et al’s) has compared the performance of RNN-based and parametric approaches such as Hawke processes, which showed that RNN-based models outperformed other alternatives. Thus, we built on top of the understandings of previous work and focused this work on how to improve RNN-based approaches for time-based sequences.
>
> Additional features such as "time of day" and "day of week" are indeed useful. However, these features are already well discretized (e.g., 24 hours a day and 7 days a week), and can be directly fed to the embedding layer. Our focus in this work is to explore a good representation for continuous time that does not have a good way for tokenization yet.
>
> Reviewer1 brought up a good point (#2) about time representation. In the revision, we added a brief discussion about the learned time representation by our TimeJoint method.

---

### Official Review · AnonReviewer2 · 2017-11-28
**Minor technical contribution, not significant gains with state of the art, misrepresents previous work**

**Rating:** 4
**Confidence:** 5

**Review:**

The authors present a model base on an RNN to predict marks and duration of events in a temporal point process. The main innovation of the paper is a new representation of a point process with duration (which could also be understood as marks), which allows them to use a "time mask", following the idea of word mask introduced by Choi et al, 2016. In Addition to the mask, the authors also propose a discretization of the duration using one hot encoding and using the event duration as a regularizer. They compare their method to several variations of their own method, two trivial baselines, and one state of the art method (RMTPP) using several real-world datasets and report small gains with respect to that state of the art method.

Overall, the technical contribution of the paper is minor, the gains in performance with respect to a single state of the art are minimal, and the authors oversell their contribution specially in comparison with the related literature. More specifically, my concerns, which prevent me from recommending acceptance, are as follows:

- The authors assume the point process contains duration and intervals, however, point processes generally do not have duration per event but they are discrete events localized in particular time points. Moreover, the duration in their representation (Figure 1) is sometimes an interevent time and sometimes a duration, which makes the whole construction inconsistent. Moreover, what happens then to the representation depicted in Figure 1 when duration is nonexistent or zero?

- The use of "time mask" is not properly justified and the authors are just extending the idea of word mask to their setting -- it is unclear why the duration of an event is going to provide context and in any case this seems like a minor technical contribution.

- The use of a time mask does not appear "more principled" than previous work (Due et al., Mei & Esiner, Xiao et al.). Previous work use the framework of temporal point processes in a principled way, the current work does not. I would encourage to authors to tone down their language.

- The regularization proposed by the authors uses a Gaussian on the "prediction error" of the duration or just cross entropy on a discretization of the duration. Given the inconsistency in the definition of the duration (sometimes it is duration, sometimes is interevent time), the resulting regularization may lead to unexpected/undesirable results. Moreover, it is unclear why the authors do not model the duration time with an appropriate distribution (e.g., Weibull) and add the log-likelihood of the durations under that distribution as regularization.

- The difference in performance  with respect to a single nontrivial baseline (the remaining baselines are trivial or versions of their own model) is minimal. Moreover, the authors fail to compare with other methods, e.g., the method by Mei & Eisner, which beats RMTPP. This is specially surprising since the authors mention such work in the related work and there is available source code at https://github.com/HMEIatJHU/neurawkes.

---

> ### Author Response · Authors · 2018-01-05
> **Responses to Reviewer2's comments**
>
> 1. R2 is right that point processes do not concern event duration. However, in many real world sequences, such as app usage, duration does exist. How long an app is used can carry important information about the nature of the event, e.g., a short versus a long YouTube watch. To deal with both duration and interval, we introduced an idle event so that both types of time spans can be represented as duration. With our framing, the nature of the “duration” depends on the event it is associated with. We feel this framing simplifies the handling of two types of time span. When duration is nonexistent in the sequence or does not matter for the domain, we don’t need to introduce idle events.
>
> 2. As discussed above, the duration of an event can provide rich information about an event. For example, in modeling app usage, a longer use of the Map Navigation app implies a more extended driving, which might lead to a different follow-up app usage. In the medical domain, the time length of a symptom is critical for predicting future symptoms and identifying the underlying cause of an illness. “time mask” is based on previous work. But it is a useful application of the contextual masking idea to continuous time that has not been explored before, which contributes new empirical evidence.
>
> 3. We have rephrased the sentence. It is not our intention to say our work is more principled. Rather, our work is focused on different aspects while previous work had a different focus. Previous work directly feeds the scalable value of time into the model. Our motivation is that because it is not clear how to properly represent time as input, we simply let the model learn the representation, similar to embeddings for words. Our approach is essentially proposing a new way to represent time instead of arguing for a completely new model. In fact, our approach can be indeed combined with previous work for better performance, e.g., using time-dependent event representation in Du’s or Mei’s model.
>
> 4. Regularization can help here because it provides additional information of next event duration in the back propagation process, which is less relevant to how duration is defined (we addressed the duration-vs-interval question above). It is important to note that we do not use Gaussian distribution to model duration. Rather we use the Gaussian distribution to model prediction errors of duration. Previous work by Hinton and Camp (COLT ’93) have discussed this approach. In fact, we do not assume any distribution for duration time in this work. Rather we hope the model will learn that from the data.
>
> 5. Our work was developed in parallel to Mei & Eisner’s work, which was discovered as an upcoming NIPS 2017 paper right before the submission. Since both Mei & Eisner’s work and RMTPP use time directly as a scalar value, we can assume our time embedding approach would bring additional accuracy to both of these methods. Again, we want to emphasize that  we are NOT proposing a new point process model but we contribute techniques that are add-on to the existing work. We look into ways to enhance event time representation with embedded times and improve training with time regularization.

---

### Official Review · AnonReviewer3 · 2017-11-29
**Approach is good**

**Rating:** 5
**Confidence:** 3

**Review:**

The paper proposes a set of methods for using temporal information in event sequence prediction. Two methods for time-dependent event representation are proposed. Also two methods for using next event duration are introduced.

The motivation of the paper is interesting and I like the approach. The proposed methods seem valid. Only concern is that the proposed methods do not outperform others much with some level of significance. More advance models may be needed.

---

> ### Author Response · Authors · 2018-01-05
> **Performance significance**
>
> We agree with the reviewer that the performance difference seems small. However, the accuracy gain from our methods, especially TimeJoint, is quite consistent across datasets. On the two of the three public datasets, the accuracy improvement is statistically significant (p<0.05). None of the time-based method seems to help on the third dataset (MIMIC). This indicates that using time might not always help. However, when it does, our methods such as TimeJoint enable more efficient representation of time than simply using the scalar value in RNN models. We have updated the paper to include these results. We also found further tuning (e.g., projection size) enabled additional performance gain for our methods, which we will add in future revisions.

---

### Author Response · Authors · 2018-01-05
**Revisions**

We updated the paper to address some of the reviewers' comments. The major changes include the following.

1. Clarified that our goal with this work is to develop time representation methods rather than a new RNN model. Our methods can enhance existing RNN models to deal with continuous time;
2. Added more experimental results, including statistical significance for the performance with three public datasets;
3. Updated Figure 3 for the performance with the three public datasets resulted from better tuning;
4. Added Figure 4 to discuss learned time representation.

---

### Decision · Program_Chairs · 2018-01-29
**ICLR 2018 Conference Acceptance Decision**

**Decision:**

Invite to Workshop Track

**Comment:**

I've summarized the pros and cons of the reviews below:

Pros:
* The method for time representation in event sequences is novel and well founded
* It shows improvements on several but not all datasets that may have real-world applications

Cons:
* Gains are somewhat small
* The task is also not of huge interest to ICLR in particular, and thus the paper might be of limited interest

As a result, because the paper is well done, but drew little excitement from any of the reviewers, I suggest that this not be accepted to the main conference, but encouraged to present at the workshop track.